# Self-consistent Gradient-like Eigen Decomposition in Solving Schrödinger Equations

## Abstract

The Schrödinger equation is at the heart of modern quantum mechanics. Since exact solutions of the ground state are typically intractable, standard approaches approximate Schrödinger's equation as forms of nonlinear generalized eigenvalue problems $F(V)V = SV\Lambda$ in which $F(V)$, the matrix to be decomposed, is a function of its own top-$k$ smallest eigenvectors $V$, leading to a "self-consistency problem". Traditional iterative methods heavily rely on high-quality initial guesses of $V$ generated via domain-specific heuristics methods based on quantum mechanics. In this work, we eliminate such a need for domain-specific heuristics by presenting a novel framework, Self-consistent Gradient-like Eigen Decomposition (SCGLED) that regards $F(V)$ as a special "online data generator", thus allows gradient-like eigendecomposition methods in streaming $k$-PCA to approach the self-consistency of the equation from scratch in an iterative way similar to online learning. With several critical numerical improvements, SC-GLED is robust to initial guesses, free of quantum-mechanism-based heuristics designs, and neat in implementation. Our experiments show that it not only can simply replace traditional heuristics-based initial guess methods with large performance advantage (achieved averagely 25x more precise than the best baseline in similar wall time), but also is capable of finding highly precise solutions independently without any traditional iterative methods.

## 1 Introduction

While the many-body Schrödinger equation governs all quantum-mechanical systems, it can be solved analytically only for extremely simple ones like an isolated hydrogen atom. Nonetheless, a series of important approximation theories are developed so that it can be solved numerically for many relevant molecules and materials including tens to thousands of electrons, making it possible for physicists and chemists to explore the properties of matters in an *ab-initio* or *first-principle* way, serving as a foundation of computational physics, chemistry and material science research.

Among those approximation theories, two mainstream directions involve the Hartree-Fock (Hartree, 1928; Hartree & Hartree, 1935) and density functional theories (Hohenberg & Kohn, 1964) that either approximate the Schrödinger equation as a "Hartree-Fock" equation or as a "Kohn-Sham equation" (Kohn & Sham, 1965). While very different in nature, both approximated equations are finally formed as a nonlinear, generalized eigenvalue problem $F(V)V = SV\Lambda$ in which $F(V)$ and $S$ are $N \times N$ real symmetric matrices, $\Lambda = \text{diag}(\lambda_1, \cdots, \lambda_k)$ is a $k \times k$ matrix containing the top-$k$ smallest eigenvalues, and $V = [v_1, \cdots, v_k]$ is an $N \times k$ matrix containing the corresponding top-$k$ eigenvectors. Interestingly, the matrix $F$ which we aim to decompose is explicitly defined as a given *function* of the eigenvectors $V$ (denoted as $F(V) : \mathbb{R}^{N \times k} \to \mathbb{R}^{N \times N}$), leading to a "cause-and-effect dilemma" or "self-consistency problem". That is, the form of the equation is defined by its final solution, but how does one know the solution if the form of the equation is not determined?

The dominant method in solve such equations is generally referred to as Self-Consistent Fields (SCFs). SCFs are the main component of nearly all mainstream quantum chemistry and solid-state physics software. The general idea is to perform a fixed-point iteration, that is, guess an initial solution $V_0$ from scratch, then solve $F(V_{k-1})V_k = SV_k\Lambda_k$ repeatedly to obtain eigenvectors $V_k$ at the $k$th iteration, $k = 1, 2, \cdots$, until $\|F(V_k) - F(V_{k-1})\|$ is less than a given convergence threshold. Unfortunately, there is no guarantee that such an iteration can always lead to a converged solution

(Froese Fischer, 1987). Actually, SCFs are extremely sensitive to the selection of initial guess $V_0$. Without a high-quality initialisation, SCFs easily fail and oscillate between two or more different results.

Till now, all popular choices for the initial guess are heuristics-based, highly reliant on specialized prior knowledge of quantum mechanics. For example, the SAD method (Van Lenthe et al., 2006) applies the physical intuition that molecular electron densities can be approximated via the sum of atomic electron densities. Here, the core Hamiltonian method ignores all interactions between electrons, which simplifies the equation to a standard eigen-decomposition problem. More methods and details can be found in (Lehtola, 2019). Besides the prerequisite of domain-specific theories, such methods are also sophisticated in implementation to achieve satisfactory convergence performance, which is a main barrier for practitioners to build high-performance solvers from scratch.

Meanwhile, in the machine learning community, researchers are facing the challenge of analyzing real-time, online data represented as streams. In contrast to the normal case that data is fully available offline, online data is received sequentially and may even change over time, thus specialized methods are developed to dynamically adapt to new patterns in the data stream, which is called stochastic, streaming or online algorithms. Specially, a series of works (Oja, 1982; Tang, 2019; Allen-Zhu & Li, 2017; Gemp et al., 2021) focus on stochastic $k$-PCA, which aim to estimate the top-$k$ principal components of a data stream in a real-time manner. To handle the stochasticity of data, such stochastic $k$-PCA methods usually contain gradient-like eigen decomposition. When a new data sample is available, the eigenvectors will be updated towards a gradient-like direction computed by the new sample, with a small learning rate.

In this work, we noticed that $F(V)$ in the aforementioned eigen-decomposition problem is subject to concurrent update during the sequential update of $V$ towards self-consistency: $V_1 \xrightarrow{F=F(V_1)} V_2 \xrightarrow{F=F(V_2)} \cdots$. In this case $F(V)$ can be regarded as a special "online data generator", which shares similarities with the online learning setting that online data is concurrently updated during the sequential learning process. In this way, we explore the possibility of applying gradient-like eigen-decomposition in stochastic $k$-PCA methods to handle the self-consistency of approximated Schrödinger equations. Together with several numerical improvements to enhance the smoothness of optimization, we developed Self-consistent Gradient-like Eigen Decomposition (SCGLED), an efficient solver for the self-consistent approximated Schrödinger equations, SCGLED is robust to initial guesses $V_0$, free of quantum-mechanism-based heuristics design, and neat in implementation. While it can simply replace traditional heuristics-based initial guess methods with performance advantage, it is also capable of finding highly precise solutions without any traditional SCF iteration.

## 2 RELATED WORK

The computational theories for quantum many-body systems, especially for determining the wave function of Schrödinger equations, has a long history starting from 1920s. There are three mainstream theories: Hartree-Fock theory (Hartree, 1928; Hartree & Hartree, 1935), density functional theory (Hohenberg & Kohn, 1964) and quantum Monte Carlo. With the prosperity of deep learning and differentiable optimization in recent years, there are a series of works focusing on deep-learning-aided wave function representation of quantum Monte Carlo methods such as FermiNet (Pfau et al., 2020) and PauliNet (Hermann et al., 2020). For density functional theory, there are also some works using neural networks to learn the exchange-correlation functional (Li et al., 2021; Kasim & Vinko, 2021). However, these works focus more on improving simulation accuracy towards physical reality by using neural networks as a better functional approximator, while our work's focus is very different, stressing on the optimization efficiency while the equation is completely given. There are also works on direct optimization for Kohn-Sham equations by total energy minimization (Yang et al., 2006), which model the problem as a constrained optimization problem instead, and more domain-specific knowledge is involved, while our work tries to solve the equation from a purely optimization-based aspect.

For gradient-like eigen-decomposition, most of the works focus on stochastic $k$-PCA, which estimates the top-$k$ principal components of a data stream. Let $x \in \mathbb{R}^d$ denote a random data sample at time step $t$, $v_i \in \mathbb{R}^d, i = 1, 2, \cdots, k$ denote the estimate of $i$th principal component (eigenvector of

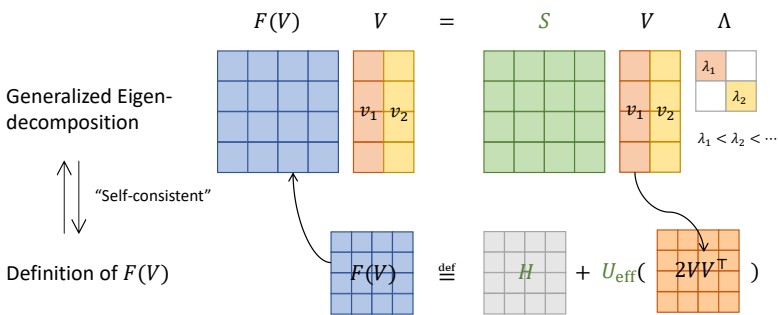

Figure 1: An visualization of the problem. $S, H \in \mathbb{R}^{N \times N}$ and $U_{\text{eff}} : \mathbb{R}^{N \times N} \to \mathbb{R}^{N \times N}$ are given input. The solution $V^*$ should obey both the generalized eigen-decomposition and the definition of $F(V)$, which leads to a "self-consistent" problem.

$\mathbb{E}[XX^\top]$) at time $t$, corresponding to the $i$th largest eigenvalue. $\eta$ denotes the learning rate.

Oja's algorithm: $\quad v_i' \leftarrow v_i + \eta(xx^\top v_i), \forall i = 1, \cdots, k \quad$ and $\quad (v_1, \cdots, v_k) \leftarrow QR(v_1', \cdots, v_k')$

Here $QR(v_1, \cdots, v_k)$ is the Gram-Schmidt decomposition that orthonormalizes $v_1^t, \cdots, v_k^t$. There are also several follow-up works (Sanger, 1989; Tang, 2019) that improve the performance, in which a most recent one is EigenGame (Gemp et al., 2021)

$$\text{EigenGame: } \nabla v_i \leftarrow 2x \left[ x^\top v_i - \sum_{j<i} \frac{\langle x^\top v_i, x^\top v_j \rangle}{\langle x^\top v_j, x^\top v_j \rangle} x^\top v_j \right], \forall i = 1, \cdots, k$$

$$\nabla^R v_i \leftarrow \nabla v_i - \langle \nabla v_i, v_i \rangle v_i, \ v_i' \leftarrow v_i + \eta \nabla^R v_i \ \text{ and } \ v_i \leftarrow \frac{v_i'}{\|v_i\|}, \forall i = 1, \cdots, k.$$

## 3 PROBLEM DESCRIPTION

In this paper, while what we want to solve is the approximated Schrödinger Equations, whose physical background is elaborated in Appendix A, we abstract it mathematically as the following type of nonlinear generalized eigenvalue problem[1]

$$F(V)V = SV\Lambda, \tag{1}$$

where

- $F(V)$: an $N \times N$ real symmetric matrix to be decomposed, which is defined in equation 2 as a function of $V$.
- $S$: an $N \times N$ positive semi-definite matrix, which is a constant input in the problem.
- $\Lambda$: $\Lambda = \text{diag}(\lambda_1, \cdots, \lambda_k)$ is a $k \times k$ diagonal matrix containing the top-$k$ smallest eigenvalues.
- $V$: $V = [v_1, \cdots, v_k]$ is an $N \times k$ matrix containing $k$ column eigenvectors corresponding to the top-$k$ smallest eigenvalues.

The definition of $F(V) : \mathbb{R}^{N \times k} \to \mathbb{R}^{N \times N}$ is as follows:

$$F(V) \stackrel{\text{def}}{=} H + U_{\text{eff}}(2VV^\top), \tag{2}$$

in which $H$ is an $N \times N$ real symmetric matrix, which is given in this problem. $U_{\text{eff}} : \mathbb{R}^{N \times N} \to \mathbb{R}^{N \times N}$ is a given function. We also define $P(V) = 2VV^\top$ for convenience. To conclude, the input

---

[1] While the regular eigenvalue problem can be described as finding $V$ that obeys $AV = V\Lambda$ in which A is an $N \times N$ matrix to be decomposed, $\Lambda = \text{diag}(\lambda_1, \cdots, \lambda_N)$ contains all eigenvalues and $V = (v_1, \cdots, v_N)$ contains the corresponding eigenvectors, a generalized eigenvalue problem adds an additional matrix $B$ with the form $AV = BV\Lambda$. When $B = I$, it degenerates to a regular eigenvalue problem.

of the problem is $S$, $H$, $U_{\text{eff}}(\cdot)$ and $k$, and the output of the problem is the eigenvectors $V^*$ that obeys both equation 1 and equation 2. The top-$k$ smallest eigenvalues of $F(V^*)$ are guaranteed to be negative.

To make it more clear, a toy example is provided as follows:

$$S = \begin{bmatrix} 1.0 & 0.6953 \\ 0.6953 & 1.0 \end{bmatrix}, H = \begin{bmatrix} -1.1204 & -0.9584 \\ -0.9584 & -1.1204 \end{bmatrix}, k = 1$$

$$[U_{\text{eff}}(P)]_{uv} = \sum_{\lambda,\sigma} P_{\lambda\sigma} E_{uv\lambda\sigma} - \frac{1}{2} \sum_{\lambda,\sigma} P_{\lambda\sigma} E_{u\lambda\sigma v},$$

$$E_{11} = \begin{bmatrix} 0.7746 & 0.4441 \\ 0.4441 & 0.5697 \end{bmatrix}, E_{12} = E_{21} = \begin{bmatrix} 0.4441 & 0.2970 \\ 0.2970 & 0.4441 \end{bmatrix}, E_{22} = \begin{bmatrix} 0.5697 & 0.4441 \\ 0.4441 & 0.7746 \end{bmatrix}$$

in which $P = 2VV^\top$, $U_{\text{eff}}(P) \in \mathbb{R}^{2\times 2}$, $E$ is a $2 \times 2 \times 2 \times 2$ tensor. The solution of the toy example should be $V^* = (0.5489, 0.5489)^\top$, in this case $F(V^*) = \begin{bmatrix} -0.3655 & -0.5939 \\ -0.5939 & -0.3655 \end{bmatrix}$, and the result of top-1 eigen-decomposition

$$\begin{bmatrix} -0.3655 & -0.5939 \\ -0.5939 & -0.3655 \end{bmatrix} V = \begin{bmatrix} 1.0 & 0.6953 \\ 0.6953 & 1.0 \end{bmatrix} V\Lambda$$

will happen to be exactly $V^* = (0.5489, 0.5489)^\top$ with the smallest eigenvalue $\lambda_1 = -0.5782$, while the other one is $[1.2115, -1.2115]^\top$ with eigenvalue $\lambda_2 = 0.6703$.

## 4 SOLVING APPROXIMATED SCHRÖDINGER EQUATIONS USING SELF-CONSISTENT GRADIENT-LIKE EIGEN DECOMPOSITION

First, notice that the stochastic $k$-PCA methods usually can be generalized to decompose a real symmetric matrix $M$. For Oja's algorithm (Oja & Karhunen, 1985), that is

$$v_i' \leftarrow v_i + \eta M v_i \quad \forall i = 1, \cdots, k \text{ and } (v_1, \cdots, v_k) \leftarrow QR(v_1', \cdots, v_k'), \tag{3}$$

in which $v_i^t$ is the eigenvector corresponding to the $i$th largest eigenvalue.

Due to the lack of efficiency, such decomposition methods are not favorable for traditional eigen-decomposition ($M$ is a given constant input) compared with classical algorithms such as QR iteration. However, as stochastic algorithms, they have a unique advantage that they can adapt to dynamic change of $M$, which suits the sticking point in equation 1 that $F(V)$ is subject to concurrent update when $V$ changes. This motivates us to apply them to tackle the approximated Schrödinger equations.

Together with the orthogonalization technique introduced in Appendix B which transforms the generalized eigenvalue problem into a standard one, we replace $M$ in equation 3 with $F' = X^\top F(V)X$, and update $F'$ in each time step immediately after each update of $V$ to maintain the self-consistency, which forms the initial version of our proposed algorithm, self-consistent gradient-like eigen decomposition (SCGLED), shown in Algorithm 1. Note that we need the eigenvectors corresponding to top-$k$ *smallest* eigenvalues rather than the largest ones, so we decompose $-F'$ instead of $F'$. Different from traditional SCF method shown in Appendix B where $V_t'$ is purely a temporary variable that should be discarded after each iteration, $V_t'$ in our proposed algorithm is more likely a "training variable" that is first randomly initialized and then trained during the whole iterative process.

While the convergence analysis of Algorithm 1 is heavily blocked by the intrinsic nonlinearity of approximated Schrodinger equations, which is fundamentally complex in computational physics, we instead use heuristics to boost the empirical convergence performance in later paragraphs, and prove the correctness of our algorithm under some robustness assumption of $F(V)$ on the convergence point, with the following proposition

**Proposition 1.** *If Algorithm 1 converges to a stable convergence point $V^*$ and for a small perturbation $\epsilon$ towards $V^*$, $[X^\top F(V^*)X - X^\top F(V^* + \epsilon)X](V^* + \epsilon) = \mathcal{O}(\epsilon^2)$, then $V^*$ is the solution of equation 1.*

The proof is deferred to Appendix C.

---

**Algorithm 1** The vanilla version of self-consistent gradient-like eigen decomposition (SCGLED)

---
**Input:** $H, S, U_{\text{eff}}(\cdot)$ in equation 1 and equation 2, learning rate $\eta > 0$
**Output:** $V^*$, the solution of equation 1
1: Initialize $V' \in \mathbb{R}^{N \times k}$ randomly.
2: Find $X$ satisfying $X^\top S X = I$.
3: **while** $V'$ is not converged **do**
4:     $V \leftarrow X V'$
5:     $F' \leftarrow X^\top F(V) X$ following equation 2
6:     Update $V'$ for one-step decomposition of $-F'$ using equation 3.
7: **end while**
8: $V^* \leftarrow X V'$

---

However, this initial version is of poor efficiency and stability. The computational efficiency is blocked by the Gram-Schmidt process which is highly order-preserving thus cannot be decentralized/vectorized. Also, due to the existence of $F(V) = H + U_{\text{eff}}(P(V))$ that is sensitive to the change of $V$, the iteration process is highly nonlinear. In this case oscillation will easily occur which hinders the iteration process toward convergence. To tackle these problems, we made several improvements to Algorithm 1 as follows.

First, we replace the classical Oja's algorithm with the decentralized version of EigenGame (Gemp et al., 2021). EigenGame replaces the time-consuming QR decomposition by a generalized Gram-Schmidt step in its gradient $\nabla v_i$, which makes it theoretically harder to analyse (since the orthonormality of eigenvectors is not guaranteed during the iteration), but with better empirical performance due to its decentralized nature. Its procedure is as follows:

$$\nabla v_i \leftarrow 2M \Big[ v_i - \sum_{j < i} \frac{v_i^\top M v_j}{v_j^\top M v_j} v_j \Big], \nabla^R v_i \leftarrow \nabla v_i - \langle \nabla v_i, v_i \rangle v_i \quad \forall i = 1, \cdots, k \tag{4}$$

$$v_i' \leftarrow v_i + \eta \nabla^R v_i, \text{ and } v_i \leftarrow \frac{v_i'}{\|v_i'\|} \quad \forall i = 1, \cdots, k \tag{5}$$

Second, to reduce oscillation during the iteration process, we introduce the damping technique for the update of $F$. That is,

$$F_t = (1 - \alpha) F_{t-1} + \alpha F(V_t), \tag{6}$$

in which $\alpha$ is the mixing hyperparameter between 0 and 1. Here we set it as 0.2.

Third, the selection of learning rate $\eta$ is highly tricky for different molecule inputs. To enhance the robustness of the algorithm towards learning rate, we introduce the momentum method for the update of $V'$ as follows:

$$m_t = \beta m_{t-1} + \eta \nabla V', \tag{7}$$
$$V' = V' + m_t, \tag{8}$$

in which the momentum term $\beta$ is set to 0.9.

Fourth, instead of updating $F$ in every iteration, we control the update interval of $F$ via a parameter $I_F$. The first reason is for efficiency, since $V'$ only update slightly in each time step with a small learning rate $\eta$, it may not be necessary to do a fresh computation of $F(V)$ in each time step, especially considering that the computation of $U_{\text{eff}}(P(V))$ is relatively time-consuming. The second reason is for stability, since the damping technique in equation 6 will degenerate if previously updated $F_{t-1}$ and current $F(V)$ are too close.

Summarizing all considerations above, our proposed algorithm is shown in Algorithm 2

## 5 EXPERIMENTS

In this section, we perform extensive performance benchmarks on W4-17 dataset(Karton et al., 2017), which is also applied in prior benchmark work (Lehtola, 2019). All the 160 singlet molecules in the W4-17 dataset are used to evaluate the performance of our proposed algorithm. Our algorithm

---

**Algorithm 2** Self-consistent gradient-like eigen decomposition (SCGLED)

---

**Input:** $H, S, U_{\text{eff}}(\cdot)$ in equation 1 and equation 2, total number of iterations $T$, learning rate $\eta$, $F$'s update ratio $I_F$
**Output:** $V^*$, the solution of equation 1
1: Initialize $V' \in \mathbb{R}^{N \times k}$ randomly.
2: $F \leftarrow H$
3: $m \leftarrow \mathbf{0}^{N \times k}$
4: Find $X$ satisfying $X^\top S X = I$.
5: **for** $t = 0, 1, 2, \cdots, T$ **do**
6:     **if** $t \bmod I_F = 0$ **then**
7:         $V \leftarrow XV'$
8:         $F \leftarrow (1 - \alpha)F + \alpha F(V)$
9:         $F' \leftarrow X^\top F X$
10:    **end if**
11:    Obtain $\nabla^R V'$ for one-step decomposition of $-F'$ using equation 4.
12:    $m \leftarrow \beta m + \eta \nabla^R V'$
13:    $V' \leftarrow V' + m_t$
14:    Normalize all column vectors in $V'$ following equation 5.
15: **end for**
16: $V^* \leftarrow XV'$

---

is implemented in Python with NumPy, and optimized via Numba (Lam et al., 2015). The learning rate $\eta$ is set to be $10^{-2}$. The effective potential matrix function $U_{\text{eff}}(\cdot)$ in Algorithm 2 is based on Hartree-Fock theory and provided by PySCF (Sun et al., 2018). We use the standard 6-31G basis set (Ditchfield et al., 1971) for the computation of all molecules. All experiments are run on a AMD Ryzen 7 5800H CPU (8 cores, 16 threads, 3.2-4.4GHz) with 16GB memory. For the reproducibility of the result, in all experiments we initialize $V'$ in Algorithm 2 by $\begin{bmatrix} I \\ 0 \end{bmatrix}$ where $I$ is a $k \times k$ identity matrix.

### 5.1 SCGLED as an initial guess method

A direct application of our proposed method is to provide an initial guess for the traditional SCF method shown in Algorithm 3. We compare with the following popular initial guess methods that are widely applied in quantum chemistry and solid-state physics software. All of them highly rely on domain-specific heuristics.

- $H_{\text{core}}$: Core Hamiltonian method, which obtains the guess orbitals from the diagonalization of the core Hamiltonian $H$, ignoring all interelectronic interactions.
- atom (Van Lenthe et al., 2006): Superposition of atomic HF density matrices.
- minao: Superposition of atomic densities projected in a minimal basis obtained from the first contracted functions in the cc-pVTZ or cc-pVTZ-PP basis set.
- huckel (Karton et al., 2017): Parameter-free Hückel guess.

In the experiment, we use PySCF's integrated implementation for all the initial guess baselines. PySCF (Python-based Simulations of Chemistry Framework, (Sun et al., 2018)) is an open-source, highly popular quantum chemistry software in Python, whose computationally critical parts are implemented and optimized in C to guarantee efficiency.

We evaluate the performance of an initial guess method from two aspects

- Precision: the distance between the initial guess and the final converged solution of equation 1, measured by the averaged energy error over 160 molecules. (expected to be as small as possible)
- Convergence: the ratio that the initial guess can successfully lead to a converged solution via traditional SCF iteration, measured by the number of molecules that failed to converge in SCF iteration with a certain initial guess method. (expected to be as few as possible)

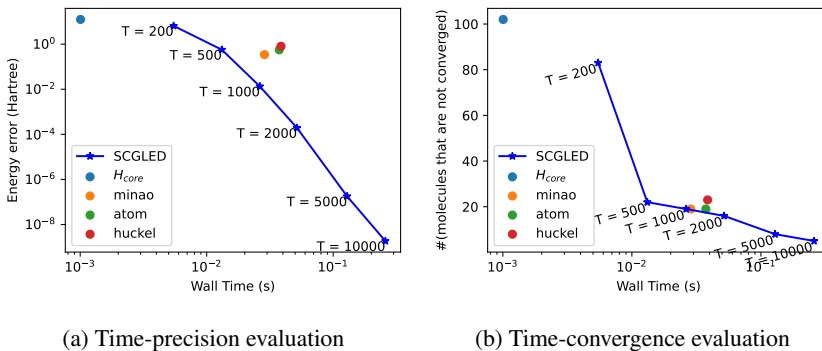

(a) Time-precision evaluation        (b) Time-convergence evaluation

Figure 2: Performance evaluation of initial guess methods

Initial guess methods are expected to be very rapid compared with main SCF iterations, so we restrict the range of $T$ from 200 to 10,000, corresponding to 5-250 milliseconds, while traditional methods take around 30 milliseconds. We also set $F$'s update interval $I_F = 50$ in this experiment.

The result of performance evaluation is shown in Figure 2. While traditional initial guess methods are parameter-free heuristics, SCGLED is iterative thus the performance and time cost are directly influenced by the number of iterations $T$. As a result, SCGLED is shown as a curve in time-performance space while traditional methods are shown as points. In both Figure 2a and Figure 2b, SCGLED's curve lies at the lower-left direction of traditional methods, indicating that our method achieves better results for both precision and convergence ability. Comparing the result of the best baseline "minao" and SCGLED at $T = 1000$ in Figure 2a, we can find that averagely SCGLED is 25x more precise than minao (0.013379 Hartree vs. 0.343194 Hartree in energy error) while the wall time are very close (26.3 ms vs. 28.6 ms). Our proposed method not only performs better or close to traditional methods given the same time, but also provides more flexible options, such as paying more time to achieve better performance in time-rich tasks, or sacrificing performance to satisfy rigorous time limitation, which is inconceivable for traditional methods due to their heuristics nature. A more detailed, per-molecule result is shown in Appendix E.

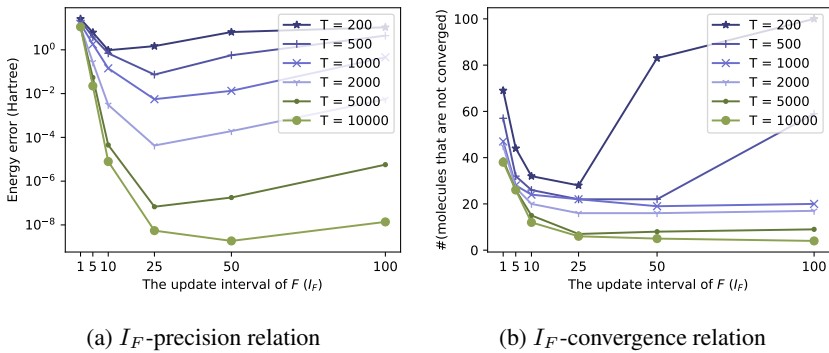

(a) $I_F$-precision relation        (b) $I_F$-convergence relation

Figure 3: $I_F$-performance relation of SCGLED

For SCGLED, we also study the influence of $I_F$, the parameter to control $F$'s update interval, to the performance, shown in Figure 3. The $I_F$-performance relation is shown as a U-shaped curve, which indicates that $I_F$ should be set to a moderate value, neither too large nor too small. The best value of $I_F$ (the bottom of the U-shaped curve) is relevant to the total number of iterations $T$: $I_F$ should be smaller for small $T$ so $F$ can be updated for reasonable times in very limited iterations, and should be moderately larger for large $T$ so that the convergence acceleration technique (such as the damping technique in equation 6) works better.

## 5.2 SCGLED AS A FULL SOLVER

While gradient-based methods are widely used in machine learning in ways that do not require extreme precision, it is usually computationally intractable in other fields that require high precision, especially scientific computing, due to its low convergence rate. However, a counterintuitive result is that our proposed gradient-like method can produce highly precise solutions of equation 1 in a reasonable number of iterations.

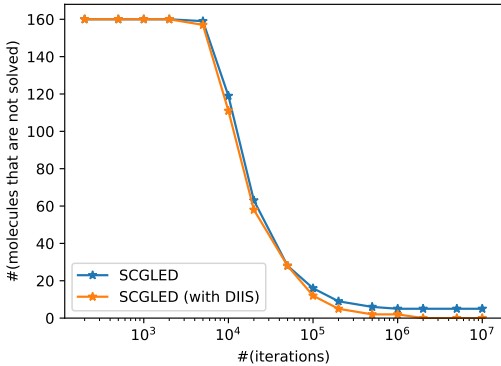

Figure 4: Performance of SCGLED for solving molecules independently.

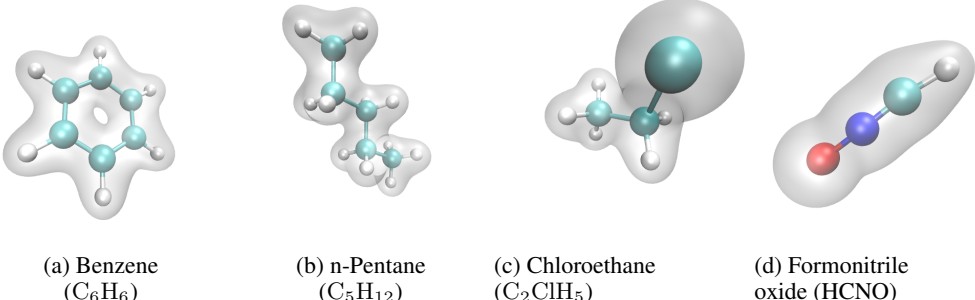

| (a) Benzene | (b) n-Pentane | (c) Chloroethane | (d) Formonitrile |
| $(C_6H_6)$ | $(C_5H_{12})$ | $(C_2ClH_5)$ | oxide (HCNO) |

Figure 5: Visualization of electron density for some molecules, solved via SCGLED.

The result that our proposed method acts as a full solver (without any traditional SCF iteration) is shown in Figure 4. The convergence criterion is selected as the default setting in PySCF. That is, the difference of a molecule's total energy before and after a single-step SCF iteration should be less than $10^{-10}$, which is extremely strict. $I_F$ is set as 100 since $T$ is relatively large in this scenario. It shows that our proposed method successfully solved more than half of the molecules within 20,000 iterations, and 155 out of 160 molecules within 1 million iterations. If we replace the simple damping technique in Algorithm 2 with a more powerful DIIS method (Pulay, 1980) for the last 10% iterations[2], we can successfully solve all 160 molecules with very high precision. Some of the highly precise solutions are visualized in Figure 5 by VMD (Humphrey et al., 1996), in which the coordinate and type of atoms are given for each molecule, and the electron densities are solved by SCGLED, represented as isosurfaces shown as grey surface around the molecule.

## 6 CONCLUSION

In this work, we solve the approximated Schrödinger equations from scratch with gradient-like eigen-decomposition. This work contributes to both the field of computational physics and machine learning as follows:

---

[2]E.g., for $T = 10000$, damping is applied for the first 9000 iterations, and we apply DIIS for the last 1000 iterations (10 actual DIIS computations are done since $I_F = 100$).

- For computational physics, beside the performance advantage, this work shows the possibility to solve the approximated Schrödinger equations from a purely optimization-based aspect, without any heuristics method based on prior quantum mechanism knowledge to bootstrap the solving stage. In this way the solving of approximated Schrödinger equations can be stripped out from its physical background, studied independently as a mathematical optimization problem.

- For machine learning, this work explores a brand new area of "self-consistent" eigenvalue problems, especially approximated Schrödinger equations, for stochastic eigendecomposition methods such as Oja's algorithm and EigenGame, which are previously regarded as specialized methods for $k$-PCA. While such methods can properly handle stochasticity, this work shows that they are also capable of handling self-consistency, which leads to a potential of application in a broader field of scientific computing.

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

## A  PHYSICAL BACKGROUND

In this section, we briefly introduce the physical background of the problem in section 3. We refer to (Szabo & Ostlund, 1996; Martin, 2004) for more details.

In computational physics, a foundation problem is to solve the Schrödinger equation of a many-body quantum system

$$H \ket{\Psi} = E \ket{\Psi}$$

where $H$ is the Hamilton operator for a system of $M$ nuclei and $N$ electrons described by their coordinates $R_A$ and $r_i$. With the Born-Oppenheimer approximation (nuclei are much heavier than electrons), we consider the electrons to be moving in the field of fixed nuclei, thus the kinetic energy of the nuclei (approximated as zero) and the repulsion between the nuclei (approximated to be constant) can be neglected. In this case we write $H$ as

$$H = \underbrace{-\sum_{i=1}^{N} \frac{1}{2} \nabla_i^2}_{\substack{\text{kinetic energy of the} \\ \text{electrons}}} \underbrace{- \sum_{i=1}^{N} \sum_{A=1}^{M} \frac{Z_A}{|r_i - R_A|}}_{\substack{\text{coulomb attraction be-} \\ \text{tween electrons and nu-} \\ \text{clei}}} + \underbrace{\sum_{i=1}^{N} \sum_{j=i+1}^{N} \frac{1}{|r_i - r_j|}}_{\substack{\text{repulsion between elec-} \\ \text{trons}}}$$

and $\Psi(r_1, \cdots, r_N)$ the electronic wave function, which should be normalized (i.e., $\langle \Psi | \Psi \rangle = \int \Psi^*(r_1, \cdots, r_N) \Psi(r_1, \cdots, r_N) dr_1 \cdots dr_N = 1$). For convenience, let operator $h(i) = -\frac{1}{2} \nabla_i^2 - \sum_{A=1}^{M} \frac{Z_A}{|r_i - R_A|}$ so $H$ can be rewritten as $H = \sum_{i=1}^{N} h(i) + \sum_{i=1}^{N} \sum_{j=i+1}^{N} \frac{1}{|r_i - r_j|}$. According to the variational principle, the problem of finding solution $\Psi$ for the ground state energy $E_0$ can be transformed to the following constrained optimization problem

$$\min \bra{\Psi} H \ket{\Psi} \quad s.t. \langle \Psi | \Psi \rangle = 1$$

Since the multi-electron wave function $\Psi(r_1, \cdots, r_N)$ is computationally intractable when $N$ is large, a basic way is to approximate it as the product of $N$ orbital wave function $\psi_1(r_1)\psi_2(r_2)\cdots\psi_N(r_N)$ satisfying $\langle \psi_i | \psi_i \rangle = \int \psi_i^*(r)\psi_i(r)dr = 1, \forall i \in 1 \cdots N$, which is called "Hartree approximation". We also expand $\psi_i(r) = \sum_{u=1}^{K} C_{ui}\phi(r)$ as a linear combination of $K$ "basis functions" which is fixed and given, in this case our task becomes to determine the value of all $C_{ui}$ so as to determine the approximated wave function. To construct a Lagrange multiplier

$$L = \bra{\Psi} H \ket{\Psi} - \sum_{i=1}^{N} \epsilon_i (\langle \psi_i | \psi_i \rangle - 1)$$

we have

$$\bra{\Psi} H \ket{\Psi} = \sum_{i=1}^{N} \int \Psi^*(r_1, \cdots, r_N) h(i) \Psi(r_1, \cdots, r_N) dr_1 \cdots dr_N$$

$$+ \int \Psi^*(r_1, \cdots, r_N) \sum_{i=1}^{N} \sum_{j=i+1}^{N} \frac{1}{|r_i - r_j|} \Psi(r_1, \cdots, r_N) dr_1 \cdots dr_N$$

$$= \sum_{i=1}^{N} \int \psi_i^*(r) h(i) \psi_i(r) dr + \frac{1}{2} \sum_{i=1}^{N} \sum_{j \neq i}^{N} \int \psi_i^*(r_i)\psi_j^*(r_j) \frac{1}{|r_i - r_j|} \psi_i(r_i)\psi_j(r_j) dr_i dr_j$$

$$= \sum_{i=1}^{N} \sum_{u,v} C_{ui} C_{vi} \int \phi_u^*(r) h(i) \phi_v(r) dr$$

$$+ \frac{1}{2} \sum_{i=1}^{N} \sum_{j \neq i}^{N} \sum_{u,v,\lambda,\sigma} C_{ui} C_{\lambda j} C_{\sigma j} C_{vi} \int \phi_u^*(r_i) \phi_\lambda^*(r_j) \frac{1}{|r_i - r_j|} \phi_\sigma(r_j) \phi_v(r_i) dr_i dr_j$$

and

$$\sum_{i=1}^{N} \epsilon_i (\langle \psi_i | \psi_i \rangle - 1) = \sum_{i=1}^{N} \epsilon_i (\int \psi_i^*(r) \psi_i(r) dr - 1)$$

$$= \sum_{i=1}^{N} \epsilon_i (\sum_{u,v} C_{ui} C_{vi} \int \phi_u^*(r) \phi_v(r) dr - 1)$$

Let $H_{uv} = \int \phi_u^*(r) h(i) \phi_v(r) dr$, $S_{uv} = \int \phi_u^*(r) \phi_v(r) dr$ and $E_{uv\lambda\sigma} = \int \phi_u^*(r_1) \phi_\lambda^*(r_2) \frac{1}{|r_1 - r_2|} \phi_\sigma(r_2) \phi_v(r_1) dr_1 dr_2$ which are all constants since $\{\phi_i\}$ are given functions, we have

$$L = \sum_{i=1}^{N} \sum_{u,v} C_{ui} C_{vi} H_{uv} + \frac{1}{2} \sum_{i=1}^{N} \sum_{j \neq i}^{N} \sum_{u,v,\lambda,\sigma} C_{ui} C_{\lambda j} C_{\sigma j} C_{vi} E_{uv\lambda\sigma}$$

$$- \sum_{i=1}^{N} \epsilon_i (\sum_{u,v} C_{ui} C_{vi} S_{uv} dr - 1)$$

$$\frac{\partial L}{\partial C_{ui}} = 2 \sum_{v} C_{vi} (H_{uv} + \sum_{j(\neq i)} \sum_{\lambda,\sigma} C_{\lambda j} C_{\sigma j} E_{uv\lambda\sigma} - \epsilon_i S_{uv})$$

Let $\frac{\partial L}{\partial C_{ui}} = 0, \forall i = 1 \cdots N, u = 1 \cdots K$ and we have

$$\sum_{v} C_{vi} (H_{uv} + \sum_{j(\neq i)} \sum_{\lambda,\sigma} C_{\lambda j} C_{\sigma j} E_{uv\lambda\sigma}) = \epsilon_i \sum_{v} C_{vi} S_{uv}, \forall i = 1 \cdots N, u = 1 \cdots K$$

Let $P_{\lambda\sigma} = \sum_{j(\neq i)} C_{\lambda j} C_{\sigma j}$, $[U_{\text{eff}}(P)]_{uv} = \sum_{\lambda,\sigma} P_{\lambda\sigma} E_{uv\lambda\sigma}$, $F_{uv} = H_{uv} + [U_{\text{eff}}(P)]_{uv}$ and $\Lambda = \text{diag}(\epsilon_1, \cdots, \epsilon_N), P, U_{\text{eff}}(P), F \in \mathbb{R}^{K \times K}$, we have the matrix form of the above equation

$$FC = SC\Lambda$$

which is a generalized eigenvalue problem where the matrix $F$ to be decomposed is defined by the eigenvectors $C$.

Actually, there are several improved approximation theories based on the Hartree approximation mentioned above, in which the definitions of $P$ and $U_{\text{eff}}(P)$ are different. A most influential one is the Hartree-Fock theory in which the multi-electron wave function is approximated by a slater determinant

$$\Psi(x_1, \cdots, x_N) = \frac{1}{\sqrt{N!}} \begin{vmatrix} \chi_1(x_1) & \chi_2(x_1) & \cdots & \chi_N(x_1) \\ \chi_1(x_2) & \chi_2(x_2) & \cdots & \chi_N(x_2) \\ \vdots & \vdots & \ddots & \vdots \\ \chi_1(x_N) & \chi_2(x_N) & \cdots & \chi_N(x_N) \end{vmatrix}$$

to conform to the antisymmetry principle $\Psi(\cdots, x_i, \cdots, x_j, \cdots) = -\Psi(\cdots, x_j, \cdots, x_i, \cdots)$ whose famous representation is Pauli exclusion principle. In this way $P_{uv} = 2 \sum_{i}^{N/2} C_{\lambda j} C_{\sigma j}$ (or in matrix form, $P = 2CC^\top$) and $[U_{\text{eff}}(P)]_{uv} = \sum_{\lambda,\sigma} P_{\lambda\sigma} E_{uv\lambda\sigma} - \frac{1}{2} \sum_{\lambda,\sigma} P_{\lambda\sigma} E_{u\lambda\sigma v}$, in which an "exchange term" $-\frac{1}{2} \sum_{\lambda,\sigma} P_{\lambda\sigma} E_{u\lambda\sigma v}$ is added.

## B  SELF-CONSISTENT FIELD (SCF) METHOD

Self-Consistent Field (SCF) is a standard method to solve equation 1. An initial density matrix $P_0$ is generated via heuristics based on prior quantum mechanism knowledge, then the generalized

eigen-decomposition problem $F(P_{t-1})V_t = SV_t\Lambda_t$ is repeatedly solved to obtain eigenvectors $V_t$ and density matrix $P_t = 2V_tV_t^\top$ at the $t$-th iterations, $t = 1, 2, \cdots$, until $|P_t - P_{t-1}|$ is less than the convergence threshold. The detailed routine is shown in Algorithm 3.

To solve the generalized eigen-decomposition problem, it should be transformed to standard form first. To achieve this, the orthogonalization technique introduced in (Szabo & Ostlund, 1996) is applied to eliminate the overlap matrix $S$. First, find a linear transformation $X$ so that $X^\top S X = I$. There are several ways to achieve this, a popular one is named "canonical orthogonalization" which lets $X = U\text{diag}(s^{-1/2})$, where $U$ and $s$ are all eigenvectors and eigenvalues of $S$. Note that all eigenvalues of $S$ are positive so there is no difficulty of taking square roots. Then let $V' = X^{-1}V$ ($V = XV'$), we have $F(V)XV' = SXV'\Lambda$. Multiply $X^\top$ on the left and we have $X^\top F(V)XV' = X^\top SXV'\Lambda$. Let $F'(V) = X^\top F(V)X$, we have

$$F'(V)V' = V'\Lambda \tag{9}$$

which is a standard eigen-decomposition problem.

---

**Algorithm 3** Self-Consistent Field (SCF) method

---

**Input:** $H, S, U_{\text{eff}}(\cdot)$ in equation 1 and equation 2. Converge threshold $\epsilon$.
**Output:** $V^*$, the solution of equation 1
  Obtain initial density matrix $P_0$ via initial guess methods (e.g., SAD or core Hamiltonian).
  Find $X$ satisfying $X^\top SX = I$.
  $t \leftarrow 0$
  **while** $|P_t - P_{t-1}| < \epsilon$ **do**
    $F_t \leftarrow H + U_{\text{eff}}(P_t)$
    $F'_t \leftarrow X^\top F_t X$
    $t \leftarrow t + 1$
    Obtain precise eigenvectors $V'_t$ of $F'_{t-1}$ corresponding to the top-$k$ smallest eigenvalues via classical eigen-decomposition methods such as QR iteration.
    $V_t \leftarrow XV'_t$
    $P_t \leftarrow 2V_tV_t^\top$
  **end while**
  $V^* \leftarrow V_t$

---

## C   PROOF

Proof of Proposition 1:

*Proof.* $V^*$ being the solution of equation 1 means that $V^*$ contains the generalized eigenvectors corresponding to the top-$k$ smallest generalized eigenvalues of $[F(V^*), S]$. In Algorithm 1, we denote the converged $V'$ in the final step as $V^{*\prime}$. With the orthogonalization technique introduced in Appendix B, $V^*$ being the solution of equation 1 is equivalent to

1. $V^{*\prime} = X^{-1}V^*$ contains the eigenvectors corresponding to the top-$k$ smallest eigenvalues of $F^{*\prime}$.

2. $F^{*\prime} = X^\top F(V^*)X$.

When $V'$ in the iteration of Algorithm 1 is equal to the converged solution $V^{*\prime}$, $F^{*\prime} = X^\top F(V^*)X$ holds. Since $V^{*\prime}$ is already converged, the one-step decomposition of $-F^{*\prime}$ will produce no update for $V^{*\prime}$. For Oja's algorithm equation 3, letting $M = -F^{*\prime}$, this means $(v_1^*, \cdots, v_k^*) = QR(v_1^* + \eta Mv_1^*, \cdots, v_k^* + \eta Mv_k^*), \eta > 0$. Note that $v_1^*, \cdots, v_k^*$ are orthonormal guaranteed by the QR decomposition. Considering that the QR decomposition is implemented via the Gram-Schmidt process, starting from $i = 1$, we have

$$\beta_1 v_1^* = v_1^* + \eta Mv_1^* \Rightarrow Mv_1^* = \frac{\beta_1 - 1}{\eta}v_1^*$$

$$\beta_2 v_2^* = v_2^* + \eta Mv_2^* - [(v_2^* + \eta Mv_2^*)^\top v_1^*]v_1^*$$

$$= v_2^* + \eta M v_2^* - [v_2^{*\top} v_1^* + \eta v_2^{*\top} M^\top v_1^*] v_1^*$$

$$= v_2^* + \eta M v_2^* - [\eta v_2^{*\top} \frac{\beta_1 - 1}{\eta} v_1^*] v_1^*$$

$$= v_2^* + \eta M v_2^* \Rightarrow M v_2^* = \frac{\beta_2 - 1}{\eta} v_2^*$$

$$\cdots$$

in which $\beta_1 = \|v_1^* + \eta M v_1^*\|, \beta_2 = \|v_2^* + \eta M v_2^* - [(v_2^* + \eta M v_2^*)^\top v_1^*] v_1^*\|, \cdots$ are normalization factors. Therefore, $v_1^*, \cdots, v_k^*$ are all eigenvectors of $-F^{*\prime}$.

To show that $v_1^*, \cdots, v_k^*$ are eigenvectors corresponding to the top-$k$ eigenvalues, we mainly follow (Oja & Karhunen, 1985) and (Hertz et al., 2018). We start from $k = 1$ to show that $v_1^*$ should correspond to the largest eigenvalue $\lambda_1$ of $-F^{*\prime}$ to be a stable convergence point.

First, for $v_1$, the QR decomposition degenerates to normalization, that is

$$F' \leftarrow X^\top F(v_1) X, \quad v_1' \leftarrow v_1 + \eta(-F') v_1, \quad v_1 \leftarrow \frac{v_1'}{\|v_1'\|}$$

Assuming $\eta$ is small enough, it can be expanded as a power series of $\eta$. By ignoring $\mathcal{O}(\eta^2)$ terms, we have

$$F' \leftarrow X^\top F(v_1) X, \quad v_1 \leftarrow v_1 + \eta \Delta v_1, \quad \Delta v_1 = (-F') v_1 - [v_1^\top (-F') v_1] v_1$$

While we have a converged solution $v_1^*$, we already know that it is an eigenvector but do not know which eigenvalue it corresponds to. In this case, we assume that after a series of iterations, $v_1$ is very close to the eigenvector corresponding to the $\alpha$th largest eigenvalue (denoted as $v^\alpha$) of the final converged matrix $-F^{*\prime} = -X^\top F(v_1^*) X$, that is

$$v_1 = v^\alpha + \epsilon$$

where $\epsilon$ is a very small perturbation vector. Since we already know that $v_1^*$ is an eigenvector, we have $v_1^* = v^\alpha$.

Let $M = -F^{*\prime} = -X^\top F(v_1^*) X$ and $E = X^\top F(v_1^*) X - X^\top F(v_1) X = -F' - M \Rightarrow -F' = M + E$. From the assumption in the proposition we have $E v_1 = \mathcal{O}(\epsilon^2)$. Then we do one-step iteration as follows:

$$\Delta v_1 = (-F') v_1 - [v_1^\top (-F') v_1] v_1$$
$$= (M + E) v_1 - [v_1^\top (M + E) v_1] v_1$$
$$= M(v^\alpha + \epsilon) - [(v^\alpha + \epsilon)^\top M(v^\alpha + \epsilon)](v^\alpha + \epsilon) + \mathcal{O}(\epsilon^2)$$
$$= (M + E)(v^\alpha + \epsilon) - [(v^\alpha + \epsilon)^\top (M + E)(v^\alpha + \epsilon)](v^\alpha + \epsilon)$$
$$= \lambda^\alpha v^\alpha + M\epsilon - (v^{\alpha\top} M v^\alpha) v^\alpha - (\epsilon^\top M v^\alpha) v^\alpha - (v^\alpha M \epsilon) v^\alpha - (v^{\alpha\top} M v^\alpha)\epsilon + \mathcal{O}(\epsilon^2)$$
$$= M\epsilon - 2\lambda^\alpha(\epsilon^\top v^\alpha) v^\alpha - \lambda^\alpha \epsilon + \mathcal{O}(\epsilon^2)$$

Now we want to analyse the direction of $\Delta v_1$ to see whether it can conquer the perturbation $\epsilon$ and converge to $v_1^* = v^\alpha$ stably. We do it by multiplying another eigenvector $v^{\beta\top}$ on the left (ignoring the $\mathcal{O}(\epsilon^2)$ term)

$$v^{\beta\top} \Delta v_1 = (M^\top v^\beta)^\top \epsilon - 2\lambda^\alpha(\epsilon^\top v^\alpha)\delta_{\alpha\beta} - \lambda^\alpha v^{\beta\top}\epsilon$$
$$= (\lambda^\beta - \lambda^\alpha) v^{\beta\top}\epsilon - 2\lambda^\alpha(\epsilon^\top v^\alpha)\delta_{\alpha\beta}$$

We notice that, if $\beta > \alpha$ (i.e., $\lambda^\beta > \lambda^\alpha$), there will always exist a direction $\epsilon$ so that both $v^{\beta\top}\epsilon$ and $v^{\beta\top}\Delta v_1$ are larger than zero. In this case $v_1$ will flip to the other eigenvector thus cannot converge to $v^\alpha$ stably. Therefore, $\alpha$ should be 1 so that $v_1^* = v^\alpha$ corresponds to the largest eigenvalue of $-F^{*\prime}$.

For top-$k$ case, take $k = 2$ as an example, notice that the Gram-Schmidt process not only normalize the eigenvector $v_2$ but also orthogonalize it towards $v_1$, so $(v_2 + \Delta v_2)^\top v_1 = (\Delta v_2)^\top v_1 = 0$, and we already know that $v_1$ will converge to the eigenvector corresponding to the largest eigenvalue. Therefore we can only select other eigenvectors out of $v^1$ to analyse the direction of $\Delta v_2$, which shows that $v_2$ will converge to the eigenvector corresponding to the second largest eigenvalue.

$$\square$$

## D   FULL-SOLVER PERFORMANCE BENCHMARK

While SCGLED is not targeted to be an extremely precise (convergence threshold less than $10^{-10}$), end-to-end solving technique due to its first-order nature, we still compare its performance toward traditional SCF methods in a full-solver setting for reference.

We test three full-solving schemes

- Independent SCGLED: applying our proposed method, SCGLED, as a full solver without any tranditional SCF methods.
- SCF + minao: applying the traditional SCF method with "minao" as the initial guess method, which performs best within traditional methods as shown in subsection 5.1, and acts as the default setting in PySCF.
- SCF + SCGLED: applying the traditional SCF method with SCGLED as the initial guess method.

with three settings for heuristics convergence acceleration techniques

- Vanilla: no convergence acceleration method is applied.
- Damping: the damping technique shown in equation 6 is applied in SCGLED or SCF iteration.
- DIIS: the DIIS technique (Pulay, 1980) is applied in independent SCGLED or SCF iteration. Note that DIIS is not applied when SCGLED is served as an initial guess method.

So excluding vanilla SCGLED in Algorithm 1 whose empirical performance is too poor for benchmarking, we have 8 test methods in total to be benchmarked. In Table 1, we evaluate these methods from two aspects

- Convergence: evaluated via the number of successfully solved molecules within the total 160 singlet molecules in W4-17 dataset.
- Efficiency: evaluated via the averaged[3] end-to-end time cost of the method along the molecules that are successfully solved.

For the parameter setting, when SCGLED is served as an initial method, we set $T = 5000$ and $I_F = 50$ for Vanilla and Damping cases, which is relatively large since their performance are more sensitive to the quality of initial guess. For the DIIS case which are relatively not so sensitive to the initial guess, we set $T = 200$ and $I_F = 25$. When SCGLED is running independently without traditional SCF iterations, the parameter setting stays the same as in subsection 5.2.

|  | Vanilla | Damping | DIIS |
|---|---|---|---|
| Independent SCGLED | — | 155 / 574.3 ms | 160 / 657.9 ms |
| SCF + minao | 141 / 332.6 ms | 159 / 244.3 ms | 160 / 160.9 ms |
| SCF + SCGLED | **149 / 190.3 ms** | **159 / 207.1 ms** | **160 / 148.7 ms** |

Table 1: Benchmark results for three full-solving schemes with three heuristics convergence acceleration settings. In each cell, the former is the number of successfully solved molecules within 160 test molecules, and the latter is the averaged time cost.

The benchmark result is shown in Table 1 and Figure 6. While independent SCGLED works in the full-solver setting, there is still a performance gap with other methods that target for full-solving performance. Instead, SCF + SCGLED works best, especially in the Vanilla case that both the convergence performance (unsolved molecules from 19 to 11) and efficiency performance (averaged time cost from 332.6 ms to 190.3 ms) are significantly improved. For the Damping and DIIS case, the performance advantages are narrowed since the heuristics convergence acceleration techniques themselves become a more dominant factor for end-to-end performance. Also note that, as an

---

[3]Note that the time cost grows exponentially towards the size of the molecule, so the distribution of time cost is quite skewed along different size of molecules. For this reason we take the logarithm of the time cost before the average.

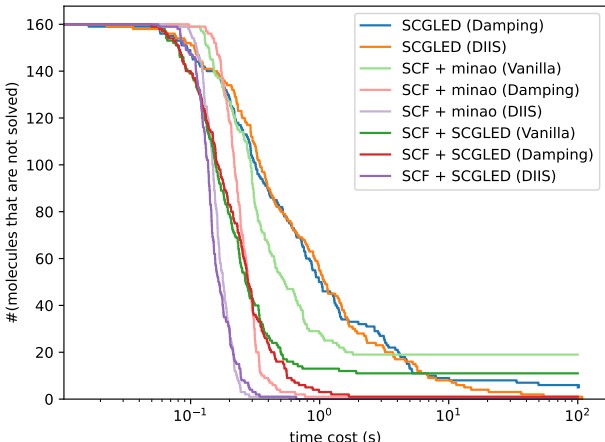

Figure 6: Full-solver Performance Benchmark.

optimization-based method, SCGLED brings more randomness compared with traditional heuristics initial guess methods that are built on domain knowledge. Therefore, while SCF + SCGLED performs better for most of the molecules, its distribution of time costs is wider, shown as curves with smaller slope in Figure 6, and the end of the curve in Damping and DIIS settings can correspond to larger time cost compared with SCF + minao.

## E  RESULT PER MOLECULE

In this section, we provide the description of the full numerical result listed in the supplementary material of this paper, for all the 160 singlet molecules in the W4-17 dataset. For each molecule and method, we list four values as follows:

- Energy: the total Hartree-Fock energy of the molecule computed with the method (Unit: Hartree, 1 Hartree = $4.3598 \times 10^{-10}$ Joule).

- Energy error: the difference between the computed energy above and the exact Hartree-Fock energy listed in the second column of the table (expected to be as small as possible). Note that we only keep 6 decimals due to the space limit, while the converge criterion is stricter, so there will be some cases that the error is $0.000000$ while the number of iterations in the fourth line is not equal to zero.

- Wall time: the wall time to run the method for the corresponding molecule.

- Convergence: whether the computed result can reach to a converged solution with the vanilla SCF iteration shown in Algorithm 3 (cannot reach a converged solution via SCF: ✗; can reach a converged solution via SCF: ✓; already a converged solution: ✓✓). If it can reach to a converged solution, we also show the number of SCF iterations it costs. If the number of SCF iterations is equal to 0, it means that the method already reaches a precise (converged) solution before the SCF iteration.

In the result table, we **bold** the result if it is a dominated result over the baselines (the energy error and wall time are both smaller than the best of the baseline methods while the number of iterations is not larger than the best baseline). We underline the result if it takes more time than the best baseline while the other target values are dominated. The $H_{\mathrm{core}}$ method are not taken into account here since both the energy error and the convergence performance are very unsatisfactory, even though it is extremely fast.

The table shows that

- 75 out of 160 molecules contain at least one dominated result (in bold).
- 155 out of 160 molecules contain at least one more precise result (marked with underline).

- 72 out of 160 molecules contain at least one result that already reached the converged solution (marked with ✓✓, the number of SCF iterations is equal to 0) within 10,000 iterations of SCGLED. That is, we already obtained the converged, precise solution. No further SCF iterations are needed.
- 14 out of 160 molecules are successfully led to convergence by SCGLED while all the four baseline methods failed to do so.

The full table of numerical result can be found in "`full_numerical_result.pdf`" as a supplementary material of this paper.

