# OpenReview forum: "Self-consistent Gradient-like Eigen Decomposition in Solving Schrödinger Equations"
_ICLR.cc/2022/Conference — ICLR 2022 Submitted_

### Official Review · Reviewer_2D5z · 2021-10-24

**Correctness:** 4
**Technical Novelty And Significance:** 4
**Empirical Novelty And Significance:** 4
**Recommendation:** 8
**Confidence:** 4

**Main Review:**

This paper is interesting that the proposed SCGLED can replace traditional heuristics-based initial guess methods with large performance advantages. I recommend this paper to be published after addressing my below comments.

My comments are as follows:

1.	For Equation (3) on Page 4, v_i’ is used, but in the description v_i_t is used. This will cause confusion.
2.	For Figure (4) on Page 8, it only displayed how fast in terms of iterations the SCGLED solves the molecules. I think it will also be interesting to show how fast in terms of absolute time the molecules are solved by 1) SCGLED, 2) SCGLED (with DIIS), and 3) traditional method


Conclusion:

This paper proposed a new method to solve approximated Schrodinger equations with gradient-like eigen-decomposition. It shows it not only can simply replace traditional heuristics-based initial guess methods with large performance advantage, but also is capable of finding highly precise solutions independently without any traditional iterative methods.


**Summary Of The Paper:**

The manuscript entitled “SELF-CONSISTENT GRADIENT-LIKE EIGEN DECOMPOSITION IN SOLVING SCHR¨O DINGER EQUATIONS” presents a new framework, Self-consistent Gradient-like Eigen Decomposition (SCGLED), to solve the approximated Schrodinger equations with gradient-like eigen-decomposition. It regards F(V) as a special “online data generator”, thus allows gradient-like eigen decomposition methods in streaming k-PCA to approach the self-consistency of the equation in an iterative way similar to online learning. SCGLED is shown to be robust to initial guesses, free of quantum-mechanism-based heuristics designs. This paper demonstrates SCGLED can replace traditional heuristics-based initial guess methods with large performance advantages and is capable of finding highly precise solutions independently without any traditional iterative methods.

**Summary Of The Review:**

In summary, I recommend its publication in ICLR after my comments are addressed.

---

> ### Author Response · Authors · 2021-11-16
> **Thank you for your constructive suggestions and positive feedback**
>
> Thank you for your constructive suggestions and positive feedback. We are glad that you think our work is interesting.
>
> We have made the following changes to the manuscript to address your comments.
>
> 1. We updated the description to be consistent with equation (3).
>
> 2. We add Appendix D to elaborate the full-solve case. We test independent SCGLED, SCF initialized by “minao” (traditional method) and SCF initialized by SCGLED (mixed method) in three different settings (vanilla, Damping and DIIS). Note that SCGLED is not targeted to be an extremely precise (convergence threshold less than $10^{-10}$), end-to-end solving technique due to its first-order nature. The result is as follows
>
> | | Vanilla | Damping | DIIS |
> |  ----  | ----  |  ----  | ----  |
> | Independent SCGLED | --- | 155 / 574.3 ms | 160 / 657.9 ms |
> | SCF + minao | 141 / 332.6 ms | 159 / 244.3 ms | 160 / 160.9 ms |
> | SCF + SCGLED | **149 / 190.3 ms** | **159 / 207.1 ms** | **160 / 148.7 ms** |
>
> In each cell, the former is the number of successfully solved molecules within 160 test molecules, and the latter is the averaged time cost. The result shows that, while independent SCGLED works in the full-solver setting, there is still a performance gap with other methods that target for full-solving performance. Instead, SCF + SCGLED works best, especially in the Vanilla case that both the convergence performance (unsolved molecules from 19 to 11) and efficiency performance (averaged time cost from 332.6 ms to 190.3 ms) are significantly improved.

---

### Official Review · Reviewer_pQwL · 2021-11-03

**Correctness:** 3
**Technical Novelty And Significance:** 2
**Empirical Novelty And Significance:** 2
**Recommendation:** 5
**Confidence:** 2

**Main Review:**

Strengths:
S1. Some experiments have shown that the proposed method outperforms the traditional heuristics-based initial guess methods in terms of accuracy and efficiency.
S2. Sufficient technical details of the gradient-like eigendecomposition algorithm are included.

Weaknesses :
S1. The authors just simply apply Oja & Karhunen's gradient method algorithm to solve their nonlinear eigenvalue problem. Such extensions seem incremental.
S2. No convergence analysis for Algorithm 1 is provided.





**Summary Of The Paper:**

This paper considers a gradient-like eigendecomposition algorithm for solving nonlinear eigenvalue problems. The proposed method can eliminate the need for domain-specific heuristics for classical methods, it is robust to initialization, free of heuristics designs, and neat in implementation. Some experiments have shown that it is more efficient and effective.



**Summary Of The Review:**

See above.

---

> ### Author Response · Authors · 2021-11-16
> **Thank you for your review**
>
> Thank you for your review. The detailed responses regarding each problem are listed below.
>
> **Q1:** “just simply apply Oja & Karhunen's gradient method algorithm to solve their nonlinear eigenvalue problem”
>
> **A1:** We hope to clarify our novelty and contribution from the following three aspects:
>
> - **The core challenge of our problem is very different: stochasticity vs. self-consistency.** We are not applying these methods to another stochastic, streaming or online scenario. Instead, our problem is perfectly deterministic and offline, and the core challenge of our problem is actually self-consistency, a cause-and-effect dilemma (the form of the equation is defined by its final solution) that is fundamental in computational physics. We surprisingly found that Oja & Karhunen's gradient methods, which are known as stochastic k-PCA methods, have the potential to handle the self-consistency issue in computational physics, and developed SCGLED following this line.
> - **Simply applying Oja & Karhunen's gradient method will simply not work.** There is a huge gap between linear and nonlinear settings. As we stated on page 4, simply applying the method (shown in Algorithm 1) will be of poor efficiency and stability. To be more precise, they can only solve toy-sized molecules such as a hydrogen molecule, and can hardly converge for even small-sized molecules such as a water molecule, not to mention the efficiency. To enhance the algorithm from such a weak status to challenging baseline methods that exploit domain knowledge and have dominated for decades, we propose four numerical improvements which can significantly boost the efficiency and stability, indispensable to the success of the algorithm.
> - **The road in computational physics is narrow.** While the machine learning community may be used to a boom of “novel” methods for a specific task every year, the computation physics is not, especially for the solving of Schrodinger equation in which traditional SCF methods with heuristics initial guess have dominated for decades. Very few numerical methods actually work from the 1930s to now, and it is unconventional enough that the heuristics part in SCF can be completely eliminated, not to mention the performance advantage.
>
> **Q2:** “No convergence analysis for Algorithm 1 is provided”
>
> **A2:** We are more than willing to provide one if such a possibility exists. However, different from linear k-PCA tasks that are thoroughly studied, intrinsic nonlinearity in Schrodinger equations is a fundamentally complex problem in computational physics that hinders us from convergence analysis. Few attempts on convergence analysis[1, 2] report failure cases in very simplified settings. Instead, we aim to prove that a stable convergence point of our algorithm is the solution. That is, assuming that our algorithm finally converged (with some heuristics design to improve its empirical convergence performance), the converged solution should be correct.
>
> References:
>
> - [1] E. Cancès and C. L. Bris, “On the convergence of SCF algorithms for the Hartree-Fock equations,” ESAIM: Mathematical Modelling and Numerical Analysis, vol. 34, no. 4, pp. 749–774, Jul. 2000, doi: 10.1051/m2an:2000102.
> - [2] C. Yang, W. Gao, and J. C. Meza, “On the Convergence of the Self-Consistent Field Iteration for a Class of Nonlinear Eigenvalue Problems,” SIAM J. Matrix Anal. Appl., vol. 30, no. 4, pp. 1773–1788, Jan. 2009, doi: 10.1137/080716293.

---

### Official Review · Reviewer_9vsL · 2021-11-03

**Correctness:** 2
**Technical Novelty And Significance:** 2
**Empirical Novelty And Significance:** 2
**Recommendation:** 5
**Confidence:** 3

**Main Review:**

Proposition 1 is confusing. The statement of the proposition itself does not make any sense. What exactly does "negligible" mean? The gradient to be 0? If so, then F = 0 everywhere and the proposition is meaningless. The proof is also hard to understand. It seems that the authors are trying to prove something along the lines of "a stable local convergence point is the solution of equation 1", but I am not sure.

Judging from the experiment results, we still need to combine the proposed method with the SCF to achieve a good solution. Can the authors give more information on the full solver case? For example, a comparison between SCF and SCGLED. How many of the examples the two algorithms fail to converge? If converge, what is the perfermance?

Some minor problems: Change the symbol to distinguish clearly $V_{eff}$ and $V$.

Equation (2) (3) can be written in the matrix form to be more consistent with the rest of the paper.


**Summary Of The Paper:**

The paper studies the Schrodinger equations and its related nonlinear generalized eigenvalue problem. It proposes the Self-consistent Graident-like Eigen Decomposition and shows that it achieves better numerical permanance for solving Schrodinger equations.

**Summary Of The Review:**

I think both the theoretical and empirical contribution of the paper is limited. I consider this paper to be marginally below the acceptance threshold.

---

> ### Author Response · Authors · 2021-11-16
> **Thank you for your constructive comments**
>
> Thank you for your constructive comments. We have made the following changes to the manuscript to address your concerns.
>
> First, we modified the statement of Proposition 1 with its proof to make it more precise, especially to clarify how “negligible” the change of F(V) should be here. The modified version is
>
> > If Algorithm 1 converges to a stable convergence point $V^*$ and for a small perturbation $\epsilon$ towards $V^*$, $[X^\top F(V^*)X - X^\top F(V^* + \epsilon)X](V^* + \epsilon) = \mathcal{O}(\epsilon^2)$, then $V^*$ is the solution of Equation 1.
>
> Then, we add Appendix D to elaborate the full-solve case. We test independent SCGLED, SCF initialized by “minao” (traditional method) and SCF initialized by SCGLED (mixed method) in three different settings. Note that SCGLED is not targeted to be an extremely precise (convergence threshold less than $10^{-10}$), end-to-end solving technique due to its first-order nature. The result is as follows
>
> | | Vanilla | Damping | DIIS |
> |  ----  | ----  |  ----  | ----  |
> | Independent SCGLED | --- | 155 / 574.3 ms | 160 / 657.9 ms |
> | SCF + minao | 141 / 332.6 ms | 159 / 244.3 ms | 160 / 160.9 ms |
> | SCF + SCGLED | **149 / 190.3 ms** | **159 / 207.1 ms** | **160 / 148.7 ms** |
>
> In each cell, the former is the number of successfully solved molecules within 160 test molecules, and the latter is the averaged time cost. The result shows that, while independent SCGLED works in the full-solver setting, there is still a performance gap with other methods that target for full-solving performance. Instead, SCF + SCGLED works best, especially in the Vanilla case that both the convergence performance (unsolved molecules from 19 to 11) and efficiency performance (averaged time cost from 332.6 ms to 190.3 ms) are significantly improved.
>
> For the minor problems, we changed the symbol of effective potential from $V_{\text{eff}}$ to $U_{\text{eff}}$. For changing equation (3) to matrix form, we have considered this but find that we also have to write equation (4) (5) to matrix form to be consistent, which may be too burdensome for readers to understand, so we keep the current form.

---

> > ### Comment · Reviewer_9vsL · 2021-11-20
> > **Reply to the authors**
> >
> > The modified condition is still confusing. Isn't it the same as $\nabla F(V^*) = 0$?

---

> > > ### Author Response · Authors · 2021-11-22
> > > **Reply to the reviewer**
> > >
> > > Maybe a simplified example help illustrate why the condition is different than $\nabla F(V^*) = \textbf{0}$. Let $V^* = \begin{pmatrix}0 \\\\ 0\end{pmatrix}$, $V = \begin{pmatrix}v_1 \\\\ v_2\end{pmatrix}$, $F(V) = \begin{pmatrix}v_2 & -v_1 \\\\ v_2 & -v_1\end{pmatrix}$ and $X = \begin{pmatrix}1 & 0 \\\\ 0 & 1\end{pmatrix}$. In this case, $\forall \epsilon \in \mathbb{R}^2$, $[X^\top F(V^*)X - X^\top F(V^* + \epsilon)X](V^* + \epsilon) = -\begin{pmatrix}\epsilon_2 & -\epsilon_1 \\\\ \epsilon_2 & -\epsilon_1\end{pmatrix}\begin{pmatrix}\epsilon_1 \\\\ \epsilon_2\end{pmatrix} = -\begin{pmatrix}\epsilon_2 \epsilon_1 - \epsilon_1 \epsilon_2 \\\\ \epsilon_2 \epsilon_1 - \epsilon_1 \epsilon_2\end{pmatrix} = \begin{pmatrix}0 \\\\ 0\end{pmatrix}$, but $\nabla F(V^*) \neq \textbf{0}$.

---

### Official Review · Reviewer_sMJ3 · 2021-11-05

**Correctness:** 3
**Technical Novelty And Significance:** 3
**Empirical Novelty And Significance:** 3
**Recommendation:** 5
**Confidence:** 3

**Main Review:**

Efficient computation of the Schr\"odinger eigenvalue problems  is a fundamental problem in computational  physics is certainly  worth to study. This paper made an interesting attempt for approximating the top $k$-smallest eigenvalues and the corresponding eigenvectors of a generalized nonlinear eigenvalue problem associated to the Hartree-Fock approximation. of the Schr\"odinger problem. However, from my perspective, the contribution of this paper is incremental and may not be sufficiently novel for ICLR. Here are some concerns and comments.

- First, it seems that the core of the proposed algorithm is the Eigengame for finding the first $k$-PCA of the linear eigenvalue problem, although there are other minor adaptions, such as introducing the damping and momentum. To me, a simple application of an existing method from others to a new problem is not novel enough for ICLR.

- Second, the authors claim that the proposed "SCGDLED is robust to initial guesses". However, this was only demonstrated in empirical results, but not justified rigorously. Can the authors prove the convergence of the proposed algorithm? Note that the Eigengame paper contains solid theoretical convergence analysis. I am wondering if any rigorous analysis can be done here.

- Another question I had is about the original Schr\"odinger problem. In the present paper, the authors seeks an  Hartree-Fock approximation of the Schr\"odinger problem and considers a finite dimensional discretization of the Hartree approximation. Can the authors comment on whether (and how) the proposed method, especially the idea of Eigengame can be extended to solving the original  (infinite dimensional) Schr\"odinger problem without resorting the Hartree approximation?

- In the statement of Proposition 1, is the assumption simply saying that  $F$ is continuous? If so, please  state it in this way. The current statement seems not precise.

- Regarding Appendix D, I do not think it is a good idea to  include a 10-page long table to show the numerical results. I would recommend the authors select some representative data and perhaps leave the rest to some open repository.

**Summary Of The Paper:**

This paper proposes a numerical algorithm for solving a class of  nonlinear generalized eigenvalue problems  that arises from the Hartree-Fock approximation of the Schr\"odinger equation. here the focus is on finding the first $k$-smallest eigenvalues and eigenvectors. The general idea is to perform  fixed point iterations (or solve a sequence of linear eigenvalue problems) in the framework of Self-Consistent Fields. To solve for the top $k$-smallest eigenvalues and eigenvectors of the linear problems, the authors replace the classical Oja’s algorithm with the recently developed EigenGame. The advantage of EigenGame is that the time-consuming QR decomposition step in Oja’s algorithm is replaced by a generalized Gram-Schmidt step in the computation of gradient updates. The efficiency of the proposed method is demonstrated in extensive numerical experiments.

**Summary Of The Review:**

This paper provides an interesting attempt for accelerating the computation of nonlinear eigenvalue problems relevant to the Schr\"odinger equation. However, the contribution of this paper is incremental and may not be sufficiently novel for ICLR. Therefore I am inclined to rejecting the submission.

---

> ### Author Response · Authors · 2021-11-16
> **Thank you for your detailed review**
>
> Thank you for your constructive comments. The detailed responses regarding each problem are listed below.
>
> **Q1:** “It seems that the core of the proposed algorithm is the Eigengame...simple application of an existing method from others to a new problem is not novel enough for ICLR”
>
> **A1:** Sorry that we may not clarify the core of our proposed method very clearly. Actually, the replacement of EigenGame is only one of the four numerical improvements of SCGLED we proposed on page 4 (improvement 1 and 3 are for efficiency; improvement 2 and 4 are for convergency). Without EigenGame (maybe back to Oja's algorithm), SCGLED can still work perfectly, but will just be a bit slower.
>
> Instead, the core of our proposed algorithm is handling self-consistency problem, a cause-and-effect dilemma that is fundamental in computational physics, with gradient-like eigen decomposition (from Oja & Karhunen's gradient method to EigenGame, commonly referred to as “stochastic k-PCA methods”). This is not a simple adaptation since the core of the problem is very different. We are not applying these methods to another stochastic, streaming or online scenario. Instead, our problem is perfectly deterministic and offline, and what we want to handle is the dilemma that the form of the equation is defined by its final solution. We surprisingly found that Oja & Karhunen's gradient methods, which are known as stochastic k-PCA methods, have the potential to handle the self-consistency issue in computational physics, and developed SCGLED following this line.
>
> Moreover, a direct application of existing methods (such as EigenGame or Oja's algorithm) can barely work here. To be more precise, they can only solve toy-sized molecules such as a hydrogen molecule, and can hardly converge for even small-sized molecules such as a water molecule, not to mention the efficiency. To enhance the algorithm from such a weak status to challenging baseline methods that exploit domain knowledge and have dominated for decades, we also propose four numerical improvements which can significantly boost the efficiency and stability together, indispensable to the success of this algorithm.
>
> **Q2:** “Can the authors prove the convergence of the proposed algorithm?”
>
> **A2:** We are more than willing to provide one if such a possibility exists. However, different from linear k-PCA tasks that are thoroughly studied, intrinsic nonlinearity in Schrodinger equations is a fundamentally complex problem in computational physics that hinders us from convergence analysis. Few attempts on convergence analysis[1, 2] report failure cases in very simplified settings. Instead, we aim to prove that a stable convergence point of our algorithm is the solution. That is, assuming that our algorithm finally converged (with some heuristics design to improve its empirical convergence performance), the converged solution should be correct.
>
> **Q3:** “Can the authors comment on whether (and how) the proposed method, especially the idea of Eigengame can be extended to solving the original (infinite dimensional) Schrodinger problem without resorting the Hartree approximation?”
>
> **A3:** It is possible to find the wave functions directly without resorting to a finite-sized basis function set. In this case we may parameterize the wave functions in another way (maybe NN) and solve the eigenfunction problem similarly to EigenGame. However, this might be a huge computational burden and may only work on toy cases. A related work can be found in [3].
>
> **Q4:** About the statement of Proposition 1
>
> **A4:** Thank you for pointing this out. We have updated the statement of Proposition 1 with its proof to make it more precise, especially to clarify how “negligible” the change of F(V) should be here. Please check the new version.
>
> **Q5:** About the table in Appendix D
>
> **A5:** Thank you for your suggestion. We moved the 10-page table of numerical results to the “supplementary material” part of this paper as an individual file.
>
> References:
> - [1] E. Cancès and C. L. Bris, “On the convergence of SCF algorithms for the Hartree-Fock equations,” ESAIM: Mathematical Modelling and Numerical Analysis, vol. 34, no. 4, pp. 749–774, Jul. 2000, doi: 10.1051/m2an:2000102.
> - [2] C. Yang, W. Gao, and J. C. Meza, “On the Convergence of the Self-Consistent Field Iteration for a Class of Nonlinear Eigenvalue Problems,” SIAM J. Matrix Anal. Appl., vol. 30, no. 4, pp. 1773–1788, Jan. 2009, doi: 10.1137/080716293.
> - [3] I. Ben-Shaul, L. Bar, and N. Sochen, “Solving the functional Eigen-Problem using Neural Networks,” arXiv:2007.10205 [cs, math, stat], Jul. 2020, Accessed: Jul. 05, 2021. [Online]. Available: http://arxiv.org/abs/2007.10205

---

### Author Response · Authors · 2021-11-23
**General Response**

Dear reviewers and AC,

We really appreciate all the four reviewers (**sMJ3**, **9vsL**, **pQwL** and **2D5z**) for their constructive comments. In this paper, we proposed SCGLED that handled the self-consistency problem, a fundamental cause-and-effect dilemma in computational physics, with gradient-like eigen decomposition, and showed its significant advantage over traditional, sophisticated quantum-mechanism-based methods that have dominated for decades.

In the discussion stage, there are two main technical concerns that are raised by more than one reviewer, and we thoroughly addressed them in the revised version.

**1. The second assumption in Proposition 1 is not precise (reviewer sMJ3 and 9vsL).**

We carefully checked Proposition 1 again and modified the statement (on page 4) with its proof (in Appendix C) to make it more precise, especially to clarify how “negligible” the change of $F(V)$ should be here. The modified version is

> If Algorithm 1 converges to a stable convergence point $V^*$ and for a small perturbation $\epsilon$ towards $V^*$, $[X^\top F(V^*)X - X^\top F(V^* + \epsilon)X](V^* + \epsilon) = \mathcal{O}(\epsilon^2)$, then $V^*$ is the solution of Equation 1.

Reviewer **9vsL** further raised a question about the difference between the second assumption and $F(V^*) = \textbf{0}$, and we provided a simplified example to show the difference.

**2. More information on the full solver case (Section 5.2) should be provided to make a comparison between SCF and SCGLED (reviewer 9vsL and 2D5z)**

We add Appendix D in the revised version to elaborate on the full-solve case. We test independent SCGLED, SCF initialized by “minao” (baseline) and SCF initialized by SCGLED (mixed method) in three different settings. Note that SCGLED is not targeted to be an extremely precise (convergence threshold less than $10^{-10}$), end-to-end solving technique due to its first-order nature. The result is as follows

| | Vanilla | Damping | DIIS |
|  ----  | ----  |  ----  | ----  |
| Independent SCGLED | --- | 155 / 574.3 ms | 160 / 657.9 ms |
| SCF + minao | 141 / 332.6 ms | 159 / 244.3 ms | 160 / 160.9 ms |
| SCF + SCGLED | **149 / 190.3 ms** | **159 / 207.1 ms** | **160 / 148.7 ms** |

In each cell, the former is the number of successfully solved molecules within 160 test molecules, and the latter is the averaged time cost. The result shows that, while independent SCGLED works in the full-solver setting, there is still a performance gap with other methods that target for full-solving performance. Instead, SCF + SCGLED works best, especially in the Vanilla case that both the convergence performance (unsolved molecules from 19 to 11) and efficiency performance (averaged time cost from 332.6 ms to 190.3 ms) are significantly improved.

We also did some minor revisions to improve the quality according to the suggestions of reviewers, including separation of numerical results (reviewer **sMJ3**) and notation changes (reviewer **9vsL** and **2D5z**).

We sincerely believe that these updates may help us better deliver the benefits of the proposed algorithm to the ICLR community.

Thank you very much,

Authors.

---

### Comment · Area_Chair_1BXT · 2021-11-27
**Concerns**

I am concerned that this would be more competently reviewed at a different venue.  It should be reviewed by researchers who have published specifically on solving Shrodinger's equation.  We do not seem to have such reviewers in the ICLR pool.  Also, a different venue would provide a more appropriate audience for the results.

I would be very interested in the author's opinions on these concerns.

---

> ### Author Response · Authors · 2021-11-29
> **Thank you for your attention**
>
> Thank you for your attention! AI for Science started to emerge as one of the important application areas for machine learning, and as part of it, we would hope this work to appear in the machine learning community such as ICLR due to the following reasons:
>
> First, we intend to introduce and tackle the quantum problem without technical jargon in physics and other specific domain knowledge, which makes it more beneficial for ML experts. While ML-aided Schrödinger equation solving might be one of the most fundamental directions in AI for science, ML practitioners are heavily blocked by the despairing technical barrier of quantum physics. All previous influential works such as PauliNet[1], FermiNet[2] and [3, 4, 5] rely heavily on quantum literature and physical intuitions thus hard to follow by broader ML communities, which might be the main reason why none of them are published in ML venues. However, our proposed method is, as far as we know, the first work that is free of such domain knowledge. I.e., the readers are not required to have any quantum theory in mind to understand both the problem and our method. This not only serves as the main feature of our method, but also provides a great opportunity for ML researchers to investigate this fundamental problem further without technical barriers. Our paper is intended to be self-contained without any technical jargon, and we are glad to find that no reviewers raise concerns about the understandability of our work.
>
> Therefore, instead of contributing our work to computational physics community that already have expertise in quantum theory, it would be more beneficial to contribute it to ML community, so as to encourage more ML researchers without such expertise to adapt our work as a start point, so that they can explore more research opportunities of ML-aided Schrödinger equation solving.
>
> Second, ML venues do include highly specialized works on scientific problems in recent years. As an emerging application area of ML, we noticed that ML venues, including ICLR, start to include AI for Science works despite their high technical barriers, especially DimeNet (ICLR 2020, [6]), JAX MD (NeurIPS 2020, [7]) and [8, 9, 10] that focus specifically on molecular quantum systems and scientific equation solving. There are also workshops [11, 12, 13] introducing scientific problems for more ML audiences, which shows the inclusiveness of ML communities. While people take it for granted when an ML technique is targeting CV/NLP scenarios, scientific scenarios should not be an exception for ML audiences.
>
> Last but not least, the novelty of our work lies in the applications of k-PCA methods to the otherwise unknown area and the specific self-consistent Eigen problem. Our work contributes to the ML community by expanding the range of gradient-like Eigen decomposition methods, which are commonly termed as “stochastic/streaming k-PCA methods” since they are regarded as specialized methods to handle the stochasticity issue in online k-PCA, to a brand-new area of self-consistency problems. The concept of self-consistency is closely related to mean-field theory (also named self-consistent field theory), which is an active topic in ML. ML and physics are closely connected with mutual concepts such as mean-field and Boltzmann machine, which leads to a potential application of our work on a boarder range of ML applications.

---

> > ### Author Response · Authors · 2021-11-29
> > **References**
> >
> > References:
> >
> > - [1] J. Hermann, Z. Schätzle, and F. Noé, “Deep-neural-network solution of the electronic Schrödinger equation,” Nature Chemistry, vol. 12, no. 10, Art. no. 10, Oct. 2020, doi: 10.1038/s41557-020-0544-y.
> > - [2] D. Pfau, J. S. Spencer, A. G. D. G. Matthews, and W. M. C. Foulkes, “Ab initio solution of the many-electron Schr\"odinger equation with deep neural networks,” Phys. Rev. Research, vol. 2, no. 3, p. 033429, Sep. 2020, doi: 10.1103/PhysRevResearch.2.033429.
> > - [3] G. Carleo and M. Troyer, “Solving the quantum many-body problem with artificial neural networks,” Science, vol. 355, no. 6325, pp. 602–606, Feb. 2017, doi: 10.1126/science.aag2302.
> > - [4] L. Li et al., “Kohn-Sham Equations as Regularizer: Building Prior Knowledge into Machine-Learned Physics,” Phys. Rev. Lett., vol. 126, no. 3, p. 036401, Jan. 2021, doi: 10.1103/PhysRevLett.126.036401.
> > - [5] M. F. Kasim and S. M. Vinko, “Learning the Exchange-Correlation Functional from Nature with Fully Differentiable Density Functional Theory,” Phys. Rev. Lett., vol. 127, no. 12, p. 126403, Sep. 2021, doi: 10.1103/PhysRevLett.127.126403.
> > - [6] J. Klicpera, J. Groß, and S. Günnemann, “Directional Message Passing for Molecular Graphs,” presented at the International Conference on Learning Representations, Sep. 2019. Accessed: Nov. 28, 2021. [Online]. Available: https://openreview.net/forum?id=B1eWbxStPH
> > - [7] S. Schoenholz and E. D. Cubuk, “JAX MD: A Framework for Differentiable Physics,” in Advances in Neural Information Processing Systems, 2020, vol. 33, pp. 11428–11441. [Online]. Available: https://proceedings.neurips.cc/paper/2020/file/83d3d4b6c9579515e1679aca8cbc8033-Paper.pdf
> > - [8] K. T. Schütt, P.-J. Kindermans, H. E. Sauceda, S. Chmiela, A. Tkatchenko, and K.-R. Müller, “SchNet: a continuous-filter convolutional neural network for modeling quantum interactions,” in Proceedings of the 31st International Conference on Neural Information Processing Systems, Red Hook, NY, USA, Dec. 2017, pp. 992–1002.
> > - [9] L. Richter, L. Sallandt, and N. Nüsken, “Solving high-dimensional parabolic PDEs using the tensor train format,” in Proceedings of the 38th International Conference on Machine Learning, Jul. 2021, pp. 8998–9009. Accessed: Nov. 29, 2021. [Online]. Available: https://proceedings.mlr.press/v139/richter21a.html
> > - [10] J. Ingraham, A. Riesselman, C. Sander, and D. Marks, “Learning Protein Structure with a Differentiable Simulator,” presented at the International Conference on Learning Representations, Sep. 2018. Accessed: Nov. 29, 2021. [Online]. Available: https://openreview.net/forum?id=Byg3y3C9Km
> > - [11] Deep Learning for Simulation (simDL), ICLR 2021 Workshop https://simdl.github.io/
> > - [12] Machine Learning for Molecules Workshop @ NeurIPS 2020, https://ml4molecules.github.io/
> > - [13] AI for Science: Mind the Gaps, A NeurIPS 2021 Workshop https://ai4sciencecommunity.github.io/

---

### Decision · Program_Chairs · 2022-01-20

**Decision:**

Reject

**Comment:**

This paper presents a numerical approach to solving the multi-body Schrodinger equation.  Three reviews give low confidence scores and the one review with high confidence, and high score, is very brief and the reviewer appears to have a weak background in this area.  My feeling is that the ICLR reviewer pool does not contain reviewers who are really competent to review this paper.  There is a large literature in the physics community on this problem and the paper should be reviewed in an appropriate venue.  This is especially true for evaluating the empirical results. If the mathematical techniques are relevant to general machine learning, and the authors want to have an impact on machine learning community, then it should be possible to give empirical results on a problem commonly used to evaluate machine learning methods at machine learning venues. Whether or not this is important for physics should be judged by physicists.  In any case, the reviews are for the most part not enthusiastic.